# Microbiota in different digestive tract of paddlefish (*Polyodon spathula*) are related to their functions

**Chengxing Long**[1]*, **Jieqi Wu**[2], **Jialin Liu**[1]

**1** Hunan University of Humanities, Science and Technology, Loudi, China, **2** Loudi Fisheries Science Research Institute, Loudi, China

* longchengxing@163.com

**Data Availability Statement:** The relevant data of this article has been submitted to the NCBI sequence read archive (accession number is PRJNA802701 https://www.ncbi.nlm.nih.gov/).

## Abstract

Paddlefish has high economic and ecological value. In this study, microbial diversity and community structure in intestine, stomach, and mouth of paddlefish were detected using high-throughput sequencing. The results showed that the diversity and richness indices decreased along the digestive tract, and significantly lower proportion of those were observed in intestine. Firmicutes, Bacteroidetes and Proteobacteria were the dominant phyla. In top 10 phyla, there was no significant difference in mouth and stomach. But compared with intestine, there were significant differences in 8 of the 10 phyla, and Firmicutes and Bacteroidetes increased significantly, while Proteobacteria decreased significantly. There was no dominant genus in mouth and stomach, but *Clostridium*_sensu_stricto_1 and uncultured_bacterium_o_*Bacteroidales* was predominant in intestine. In conclusion, the species and abundance of microbiota in the mouth and stomach of paddlefish were mostly the same, but significantly different from those in intestine. Moreover, there was enrichment of the dominant bacteria in intestine.

## Introduction

The digestive tract is a very important and specialized organ system, mainly involved in the digestion and absorption of substances, and is also the largest immune organ of body [1]. There is a diverse and variety of microorganisms inhabited in the cavity of the digestive tract, which is responsible for maintain the physiological homeostasis of the host [2]. Therefore, understanding the characteristics of gastrointestinal microbiota is vital for host health [3]. In recent years, microbiota, especially in intestine, has become a research hotspot. Increasingly, research has revealed the relationship between different intestinal microbiota and body diseases, and predict or determine the severity of diseases based on the species and abundance in intestinal microbiota [4,5].

The microbial composition in the digestive tract is mainly affected by many factors, such as dietary, environmental and physiological factors of the host, etc. Different sections of the digestive tract exhibit various environmental conditions [6–8]. Environmental conditions can

**Funding:** The research was supported by the Natural Science Foundation of Hunan Province (2022JJ30316) to CX. The funder had no role in study design, data collection and analysis, decision to publish, or preparation of the manuscript.

**Competing interests:** The authors have declared that no competing interests exist.

impact the microbial composition in the digestive tract. In turn, the study on the characteristics of microbiota in the digestive tract can reflect the living environment [9]. Therefore, a balanced microbiota is essential for the maintenance of overall health in fish [10]. During the past decade, a large number of studies have been reported on gastrointestinal microorganisms of aquatic animals [3,11]. Studies have shown that the stability of gastrointestinal microorganisms in aquatic animals depends on the number and abundance of species, as well as complex interactions within the community [12].

Paddlefish, *Polyodon spathula*, is a large freshwater fish native to the Mississippi River basin in North America, belonging to the phylum vertebrata, class bony fish, order sturgeons, family sturgeons [13]. As an ancient freshwater economic fish, *Polyodon spathula* has strong adaptability, rapid growth, short food chain, and mainly feeds on crustacean zooplankton, and is the only species that filters and feeds zooplankton in paddlefish [14]. Paddlefish eggs can be made into caviar, is a rare tonic, meat is rich in a variety of amino acids. The content of cartilage is high, which contains high crude protein and sodium, potassium, calcium and other trace elements, has a high edible and economic value. In addition, its unique long-snout shape makes it has high ornamental value [15,16]. Therefore, paddlefish breeding has very high economic value. Furthermore, paddlefish has the physiological function of optimizing water quality because of its dense and slender gill rake [17]. In this respect, paddlefish has very good ecological value again.

The paddlefish is an important filter feeding fish in mixed pond in China. At present, many scholars have reported the intestinal microbial composition of paddlefish [18–20], but have mainly focused on breeding and reproductive technology [21–23], digestive enzymes and nutritional meat quality evaluation [14,16,24,25], probiotics [26], and involving different parts of the digestive tract [1]. In this study, hybrid aquaculture pond samples from a regional fishery science research institute were selected, and the characteristics and differences of the microbiota in the paddlefish mouth, stomach and intestine were analyzed by using second-generation high-throughput sequencing technology and bioinformatics methods. This study aims to analyze the characteristics and differences of microbiota in different parts of the paddlefish digestive tract, and to reveal their correlations with functions. The results will provide a microbiological basis for promoting the healthy development of aquaculture.

## Materials and methods

### Experimental fish and farming patterns

The paddlefish used in the experiment were collected in the same pond at the Fisheries Research Institute, Loudi, Hunan Province, China. It was the same batch of breeding fish, healthy and disease-free, with body weight 1967.2–2471.4 g. The breeding time of the fish was from Janunary 4, 2020 to Janunary 19, 2021, a total of 380 days, and the stocking specification was 350–400 g. During the whole experiment, the fish have been raised in natural water without feeding under the following water conditions: depth 2.0 m, temperature 17.8 ℃, dissolved oxygen > 4.35 mg/L, pH 8.10–8.56.

### Sample collection and processing

The sampling time was 8:00 a.m on January 19, 2021. From 21 fish caught in a trawling net, five healthy individuals with similar body weight (length 75.4–80.2 cm) were randomly selected and transported back to the laboratory in a special fish bucket, while the rest were returned to the pond.

After measuring the body length and mass of the sample fish in the laboratory, the paddlefish was deeply anesthetized in water containing overdose of MS222, the surface of the

paddlefish was rinsed with sterile water and 75% ethanol [27]. Five mouth samples (YZ1-YZ5), five stomach samples (YW1-YW5) and five intestine samples (YC1-YC5) were collected aseptically with tweezers, and stored with sterilized centrifuge tubes and refrigerated at -80 ℃ for later use.

## 16S rRNA gene sequencing

MN NucleoSpin 96 Soi kit was used extract DNA from the collected digesta samples. The V3-V4 variable region of 16S rRNA was amplified with 338F and 806R primers and followed by high-throughput sequencing using the Illumina HiSeq 2500 platform. The amplification primers, reaction system and amplification conditions are as follows. The primers were synthesized by Beijing Biomarker Technologies Co., Ltd (Beijing, China).

The sequence for the forward primer was 5′-ACTCCTACGGGAGGCAGCA-3′, and the reverse primer sequence was 5′-GGACTACHVGGGTWTCTAAT-3′. The amplification reaction was performed as follows: 5 μL KOD FX Neo Buffer, 0.3 μL (10 μM) of forward primer, 0.3 μL (10 μM) of reverse primer, 2 μL (2mM) dNTP, 0.2 μL KOD FX Neo, 50 ng of DNA template. The amplification conditions were as follows: initial denaturation at 95˚C for 5 min, followed by 25 cycles consisting of denaturation at 95˚C for 30 s, annealing at 50˚C for 30 s, extension at 72˚C for 40 s, and a final extension of 7 min at 72˚C.

## Analysis of microbial composition

Splicing the original data (FLASH, version1.2.11) [28], high-quality Tags sequence can be obtained by quality filtering (Trimmomatic, version 0.33) [29] and removal of chimera (UNHIME, version8.1) [30]. The sequences were clustered at the 97% similarity level (USEARCH, Version10.0) [31], and 0.005% of the sequence numbers was used as the threshold to filter out [32]. Mothur software (Version V.1.30, http://www.mothur.org/) was used to calculate the alpha diversity (including Chao1 richness estimator, Ace richness estimator, Shannon diversity index and Simpson Diversity index) and Beta diversity (including principal component analysis, principal coordinate analysis and non-metric multidimensional scale) in samples, respectively, so as to comprehensively evaluate the overall diversity and reveal differences among samples. LefSe analysis (http://huttenhower.sph.harvard.edu/lefse/) was used to find species with significant differences among all groups. Based on the four distance matrices obtained from beta diversity analysis, the samples were hierarchically clustered using the unweighted paired average method (UPGMA) with R to determine the similarity of species composition among the samples.

## Statistical analysis

SPSS24.0 statistical software (IBM Corp., Armonk, NY, USA) was used for data statistics. Measurement data were expressed as the means ± standard deviations (x ± s), and independent sample T test was used to compare the pairwise means. A one-way analysis of variance was used for differences among normal distribution data groups, and Mann-Whitney U test was used for non-normal distribution data. $P < 0.05$ was regarded as a significant difference. All raw data obtained in this study have been submitted to the NCBI sequence read archive (accession number is PRJNA753213 https://www.ncbi.nlm.nih.gov/).

## Ethics statement

The animal study was reviewed and approved by The Institutional Animal Care and Use Committed of Hunan University of Chinese Medicine (SYXK (Xiang) 2019–0009).

**Table 1. Coverage and diversity indices of bacterial species in paddlefish (*Polyodon spathula*).**

| group | Chao1 | Ace | Simpson | Shannon | Coverage |
|---|---|---|---|---|---|
| YC | 675.1619±88.1881 | 695.4459±88.4495 | 0.6992±0.0668 | 2.7510±0.2822 | 0.9979±0.0001 |
| YW | 996.6823±62.4031** | 1023.4613±83.1413** | 0.9723±0.0250** | 7.5987±0.8204** | 0.9977±0.0002 |
| YZ | 1049.0519±82.2196** | 1079.5702±154.4841** | 0.9758±0.0379** | 7.9248±0.9886** | 0.9977±0.0005 |

Note: YC stands for Paddlefish intestine, YW stands for paddlefish stomach; YZ stands for Paddlefish mouth; Compared with Paddlefish intestine

*stands for $p < 0.05$

**stands for $p < 0.01$.

## Results

### Sequencing characteristics of 16S rRNA gene

After quality control, a total of 1,189,396 high-quality sequences were obtained from 15 samples in mouth (YZ), stomach (YW) and intestine (YC) of paddlefish, and the proportion of effective sequences in each sample was between 0.9153 and 0.9770. The average Coverage index of each sample was 0.9978, between 0.9970 and 0.9982, which could reflect the real situation of species in the community (Table 1).

Chao1 index, Ace index, Shannon index and Simpson index were calculated to illustrate the diversity and richness of each sample. According to the calculation results of diversity and richness index (Table 1), Chao1 index, Ace index, Simpson index and Shannon index in YC presented the lowest value, all of which were significantly different from those in YZ and YW ($P < 0.01$). However, Chao1, Ace, Simpson and Shannon diversity index were similar between YZ and YW, with no significant difference. These results indicated that the lowest richness and diversity of bacteria were found in YC and the highest one in YZ, the bacteria in YZ and YW were relatively stable.

### Overall microbiota structures

A total of 35 phyla, 733 genera and 821 species were identified in all the samples from the digestive tract of the paddlefish. Among them, 25, 34 and 35 phyla, 462, 659 and 644 genera and 512, 737 and 721 species were identified in YC, YW and YZ, respectively. Among the identified phyla, bacteria from Firmicutes, Bacteroidetes and Proteobacteria took dominant. The relative abundance of these three phyla accounted for 74%, 68.45% and 92.96% of YZ, YW and YC, respectively (Fig 1).

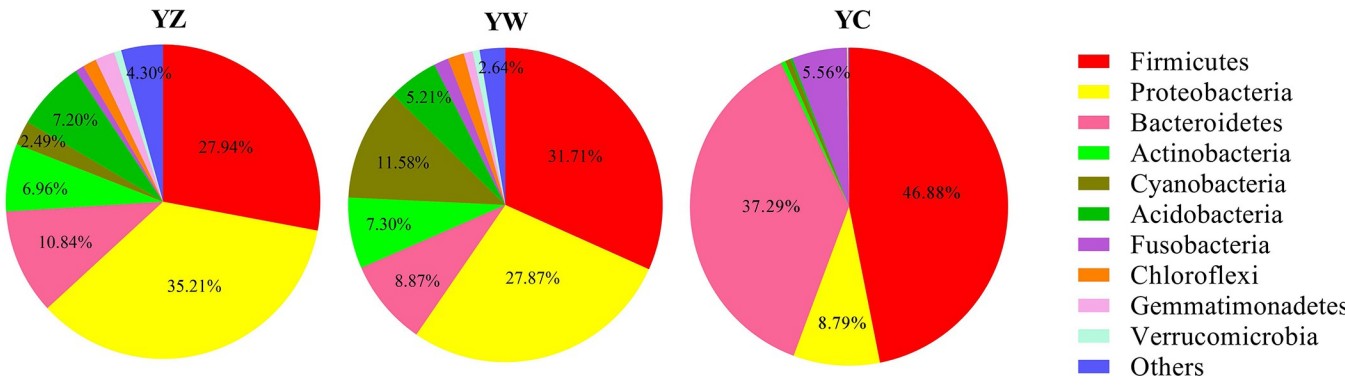

**Fig 1. The bacterial community in all samples at phylum level.** Note: YC stand for Paddlefish intestine, YW stand for paddlefish stomach; YZ stand for Paddlefish mouth.

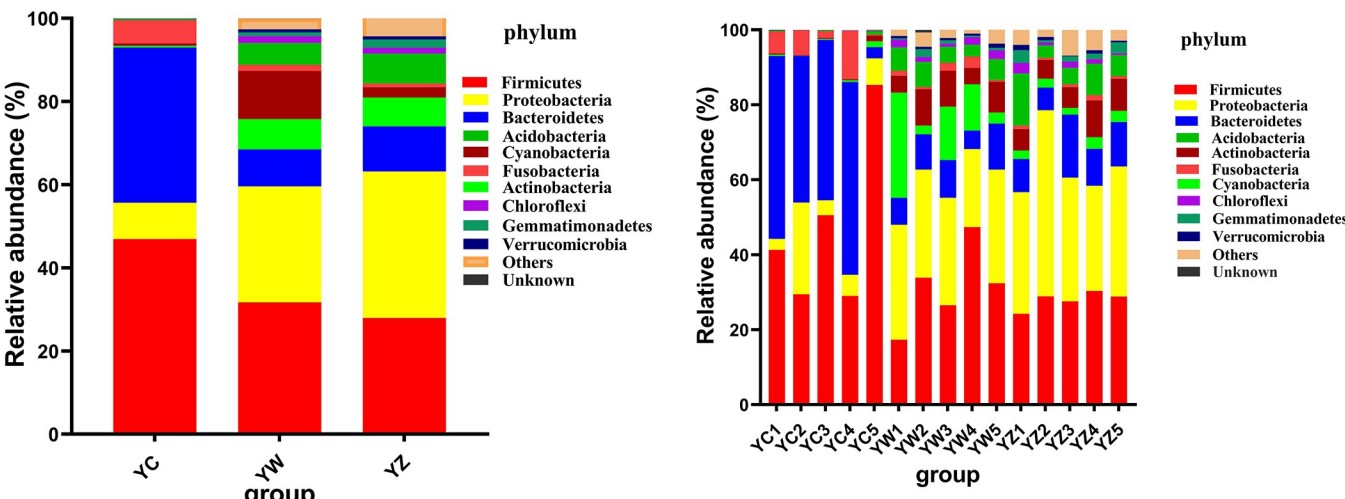

**Fig 2. Relative abundance of bacteria in each digestive tract on top 10 phyla.** Note: YC stand for Paddlefish intestine, YW stand for paddlefish stomach; YZ stand for Paddlefish mouth.

## Characteristics of bacterial community composition

**Characteristics of bacterial community composition at the phylum level.** The microbiota compositions in YC, YW and YZ were detected at the phylum level. There were 35 phyla detected in 15 samples of the three groups, including 25 in YC sample, 34 in YW sample and 35 in YZ sample. Proteobacteria (35.21%, 27.87% and 8.79% respectively), Firmicutes (27.94%, 31.71% and 46.88% respectively), Bacteroidetes (10.84%, 8.87% and 37.29% respectively), Acidobacteria (7.20%, 5.21% and 0.24% respectively), Actinobacteria (6.96%, 7.30% and 0.46% respectively) and Cyanobacteria (2.49%, 11.58% and 0.55% respectively) were the dominant phylum in YC, YW and YZ of paddlefish. However, in the top 10 phyla, the proportion of Proteobacteria, Bacteroidetes, Actinobacteria, Cyanobacteria, Acidobacteria, Chloroflexi, Gemmatimonadetes and Verrucomicrobia in YW and YZ were significantly different from YC ($P < 0.01$ or $P < 0.05$), but there was no significant difference in the abundance of these phyla in YW and YZ (Fig 2, Table 2). In addition, except for the relative abundance of Firmicutes,

**Table 2. Relative abundance of bacteria in each digestive tract of paddlefish (*Polyodon spathula*) on top 10 phyla.**

| Phylum | YC | YW | YZ |
|---|---|---|---|
| Firmicutes | 0.4710±0.2315 | 0.3148±0.1099 | 0.2796±0.0231 |
| Proteobacteria | 0.0882±0.0889 | 0.2787±0.0404** | 0.3557±0.0824** |
| Bacteroidetes | 0.3705±0.1965 | 0.0879±0.0281* | 0.1070±0.0401* |
| Actinobacteria | 0.0046±0.0055 | 0.0725±0.0265** | 0.0691±0.0210** |
| Cyanobacteria | 0.0055±0.0061 | 0.1198±0.1047* | 0.0249±0.0055** |
| Acidobacteria | 0.0024±0.0030 | 0.0521±0.0148** | 0.0713±0.0420** |
| Fusobacteria | 0.0554±0.0489 | 0.0152±0.0110 | 0.0089±0.0034 |
| Chloroflexi | 0.0004±0.0003 | 0.0168±0.0066** | 0.0138±0.0093* |
| Gemmatimonadetes | 0.0002±0.0002 | 0.0090±0.0069* | 0.0197±0.0113* |
| Verrucomicrobia | 0.0008±0.0003 | 0.0071±0.0022** | 0.0075±0.0045** |

Note: YC stands for Paddlefish intestine, YW stands for paddlefish stomach; YZ stands for Paddlefish mouth; Compared with Paddlefish intestine

*stands for $p<0.05$

**stands for $p<0.01$.

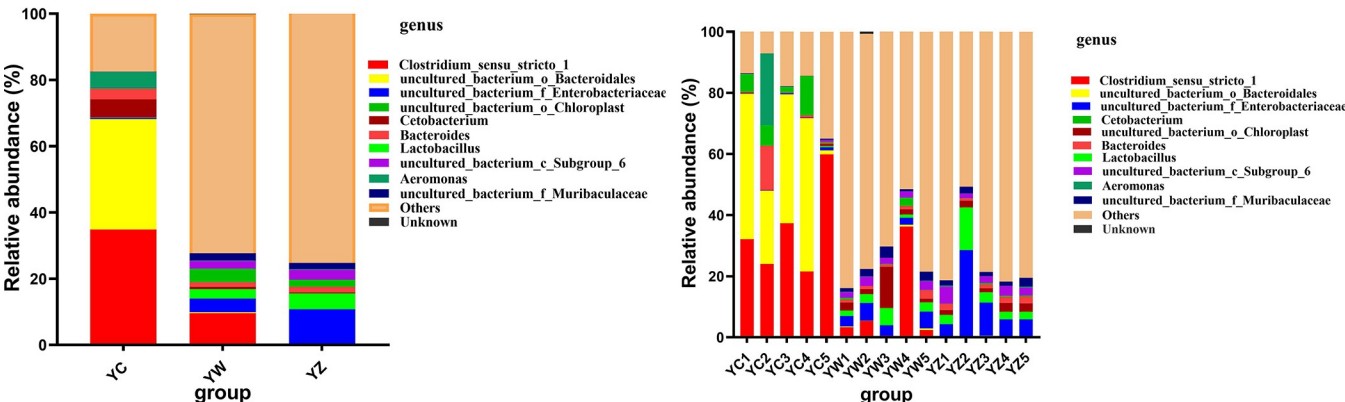

**Fig 3. Relative abundance of bacteria in each digestive tract on top 10 genus.** Note: YC stand for Paddlefish intestine, YW stand for paddlefish stomach; YZ stand for Paddlefish mouth.

Bacteroidetes and Fusobacteria increased in YC, the other seven phyla all decreased in different degrees.

## Characteristics of bacterial community composition at genus level

There were 733 genera detected in 15 samples of the three groups, including 242 uncultured bacteria. Of the 733 identified genera, 462 genera were detected in YC, 659 genera were detected in YW and 644 genera were detected in YZ of paddlefish. Furthermore, in the top 10 genera, uncultured bacteria accounted for 5. *Clostridium*_sensu_stricto_1 (34.83%) and uncultured_bacterium_o_*Bacteroidales* (33.31%) were predominant in YC, but their relative abundance was very low in YW and YZ, accounted for 9.57%, 0.16% and 0.29%, 0.14% respectively. Interestingly, there was no dominant genus in YW and YZ, and the relative abundance of the top 10 genera accounted for less than 30%. Overall, the species in YW and YZ were much richer than those in YC, but there was no dominant genus. Furthmore, there was no significant difference in the top 10 genera. These observations suggest that the microbial composition between YW and YZ was similar, and there were no significant differences (Fig 3, Table 3). Furthermore, we can get the same conclusion in UPGMA analysis (Fig 4).

**Table 3. Relative abundance of bacteria in each digestive tract of paddlefish (*Polyodon spathula*) on top 10 genus.**

| genus | YC | YW | YZ |
|---|---|---|---|
| *Clostridium*_sensu_stricto_1 | 0.3497±0.1527 | 0.0959±0.1501* | 0.0016±0.0011** |
| o_*Bacteroidales* | 0.3306±0.2047 | 0.0029±0.0026* | 0.0014±0.0014* |
| f_*Enterobacteriaceae* | 0.0037±0.0039 | 0.0407±0.0152** | 0.1089±0.1001* |
| *Lactobacillus* | 0.0013±0.0017 | 0.0289±0.0174** | 0.0508±0.0498** |
| *Cetobacterium* | 0.0022±0.0027 | 0.0410±0.0526* | 0.0215±0.0072** |
| o_*Chloroplast* | 0.0549±0.0491 | 0.0072±0.0114 | 0.0023±0.0017 |
| *Bacteroides* | 0.0317±0.0637 | 0.0134±0.0084 | 0.0167±0.0061 |
| c_*Subgroup_6* | 0.0017±0.0024 | 0.0236±0.0059** | 0.0301±0.0158** |
| *Aeromonas* | 0.0473±0.1053 | 0.0010±0.0006 | 0.0017±0.0013 |
| f_*Muribaculaceae* | 0.0011±0.0018 | 0.0216±0.0125* | 0.0193±0.0067** |

Note: YC stands for Paddlefish intestine, YW stands for paddlefish stomach; YZ stands for Paddlefish mouth; Compared with Paddlefish intestine

*stands for $p<0.05$

**stands for $p<0.01$.

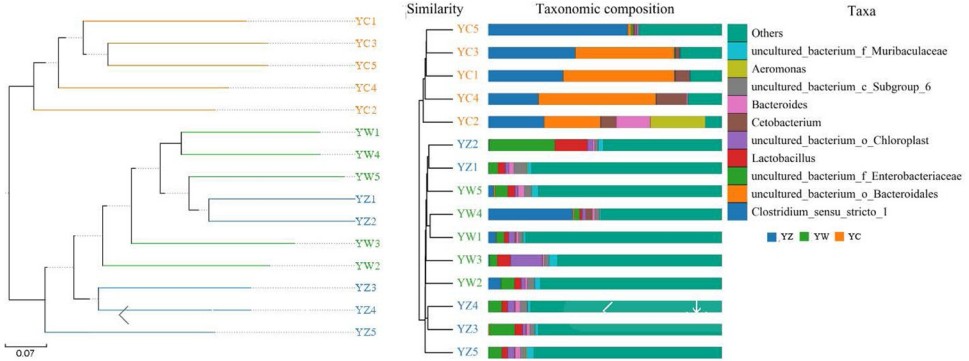

**Fig 4. Uphma analysis of bacteria in each digestive tract of Polyodon spathula.** Note: YC stand for Paddlefish intestine, YW stand for paddlefish stomach; YZ stand for Paddlefish mouth.

## Discussion

The microbial community is a complicated ecosystem, which is kept in dynamic balance by the interactions between species [33]. The balance of intestinal bacteria in healthy fish plays an important role in immune regulation and nutrient metabolism [34,35]. In the aquaculture water environment, Firmicutes, Bacteroidetes and Proteobacteria are the main microbiota [36,37]. Sustainable development of aquaculture requires full consideration of the interaction between the environment and aquatic organisms [38–40]. Therefore, good feedback can be obtained on the microbial community of the aquaculture water by analyzing the characteristics of the gastrointestinal microbial community of its colonized aquatic organisms. In this study, we investigated the characteristics of microbial community in YC, YW and YZ of paddlefish using Miseq sequencing technology and bioinformatics, and analyzed the differences among them.

From the results of Alpha diversity, the microbial diversity in YC was significantly lower than that in YZ and YW, which was consistent with the highest microbial diversity in the stomach and the lowest microbial diversity in the intestine (Yang et al., 2020). It was also consistent with the fact that the microbial diversity in the aquaculture water was significantly higher than that in the intestine [4]. Paddlefish are filter feeders, their mouths are in direct contact with the aquatic environment, and their intestines passing through the selection barrier of the stomach, thus reducing diversity [41]. These results indicate that microorganisms maintain the relative stability of community through interspecies interactions to adapt to different physicochemical conditions and functions in different organs of the body.

At the phylum level, we identified 35 phyla. The dominant phyla in YZ and YW samples were Firmicutes, Proteobacteria, Bacteroidetes, Actinobacteria, Cyanobacteria and Acidobacteria. Whereas YC was rich in Firmicutes, Bacteroidetes, Proteobacteria and Fusobacteria. Specially, the relative abundance of Firmicutes and Bacteroidetes was absolutely predominant in YC. There was no significant difference between YZ and YW in the top 10 phyla. However, compared with YC, there were significant differences in 8 of the 10 phyla, among which Firmicutes and Bacteroidetes increased significantly, while Proteobacteria decreased significantly. In general, in human intestinal microbes, there is a significant correlation between the proportion of Bacteroidetes and Firmicutes, and the ratio of Firmicutes to Bacteroidetes can reflect the lipid situation of the body [42]. It has been reported that increased members of Firmicutes can increase the amount of lipids [43]. Bacteroides is a common bacteria in the intestinal tract of aquatic fish [44,45]. The significant increase in the relative abundance of bacteroides in the

intestine can promote the digestion of carbohydrates, which is closely related to the digestive and absorption function of the intestine.

At the genus level, we identified 733 genera, including 462 in YC sample, 659 and 644 in YW and YZ samples, respectively. Of these, *Clostridium*_sensu_stricto_1 and uncultured_bacterium_o_*Bacteroidales* were in absolutely predominant in YC, but low in YW and YZ. More interestingly, of the top 10 genera, there was no dominant genus in YW and YZ. *Clostridium*_sensu_stricto_1 is the dominant genus in *Clostridiales* and belongs to Firmicutes. *Clostridium* can antagonize a variety of pathogenic fungi and also promote nitrogen accumulation [46], which indicates that the paddlefish has the potential function of optimizing water quality. *Bacteroidales* are the main representative of intestinal anaerobic microorganisms and an important microorganism that affects animal metabolism [47–50]. They play a crucial role in nutrient absorption, immune response, fat accumulation and intestinal microbiota balance, which directly or indirectly maintain the health of the host [50]. These effects directly or indirectly indicate that the composition and function of microbiota in the intestine are significantly different from those in the mouth and stomach, and these differences are related to the intestinal function.

In summary, our work confirmed the similarity of microbial species in YW and YZ from the community characteristics in YZ, YW and YC of paddlefish. This study showed that the characteristics of the microbiota community in YC were different from those in YW and YZ, and there was a phenomenon of enrichment of dominant bacterial species. These findings will further support the correlation between the colonization of specific microbiota and their functions in paddlefish.

## Author Contributions

**Conceptualization:** Jialin Liu.

**Data curation:** Jieqi Wu.

**Funding acquisition:** Chengxing Long.

**Methodology:** Jialin Liu.

**Software:** Chengxing Long.

**Writing – original draft:** Chengxing Long.

**Writing – review & editing:** Chengxing Long, Jieqi Wu, Jialin Liu.

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
