## [Decision Letter · Decision Letter 0]

14 Jan 2024

PONE-D-23-38263Microbiota in different compartments of the digestive tract of paddlefish (Polyodon spathula) are related to their functionsPLOS ONE

Dear Dr. Long,

Thank you for submitting your manuscript to PLOS ONE. After careful consideration, we feel that it has merit but does not fully meet PLOS ONE’s publication criteria as it currently stands. Therefore, we invite you to submit a revised version of the manuscript that addresses the points raised during the review process.

We look forward to receiving your revised manuscript.

Kind regards,

Nafiu Bala Sanda, PhD

Academic Editor

PLOS ONE

Journal Requirements:

We noticed you have some minor occurrence of overlapping text with the following previous publication(s), which needs to be addressed:

https://onlinelibrary.wiley.com/doi/10.1111/are.15506

https://www.sciencedirect.com/science/article/pii/S2352513420301241?via%3Dihub

In your revision ensure you cite all your sources (including your own works), and quote or rephrase any duplicated text outside the methods section. Further consideration is dependent on these concerns being addressed.

"the Natural Science Foundation of Hunan Province (2022JJ30316)."

"This work was supported by the Natural Science Foundation of Hunan Province (2022JJ30316)"

"the Natural Science Foundation of Hunan Province (2022JJ30316)."

6. Please remove your figures from within your manuscript file, leaving only the individual TIFF/EPS image files, uploaded separately. These will be automatically included in the reviewers’ PDF.

Additional Editor Comments:

There are important points raised by all the reviewers with regards to the suitability of manuscript to be publish in PLOSONE in it's current state, thus, please carefully follow the comments and responded accordingly to improve the manuscript.

Reviewers' comments:

Reviewer's Responses to Questions

**Comments to the Author**

1. Is the manuscript technically sound, and do the data support the conclusions?

Reviewer #1: Yes

Reviewer #2: No

Reviewer #3: Yes

2. Has the statistical analysis been performed appropriately and rigorously? 

Reviewer #1: Yes

Reviewer #2: No

Reviewer #3: Yes

3. Have the authors made all data underlying the findings in their manuscript fully available?

Reviewer #1: Yes

Reviewer #2: Yes

Reviewer #3: Yes

4. Is the manuscript presented in an intelligible fashion and written in standard English?

Reviewer #1: Yes

Reviewer #2: No

Reviewer #3: Yes

5. Review Comments to the Author

Reviewer #1: This study used high-throughput sequencing to analyze the microbial communities in different digestive tract compartments (mouth, stomach, and intestine) of paddlefish (Polyodon spathula). The aim is to reveal their characteristics and differences, exploring their associations with host functions. The research, focusing on the economically and ecologically valuable freshwater fish, paddlefish, holds significance for the sustainable development of aquaculture.

Reviewer's Comments:

1. Provide more detailed descriptions of the high-throughput sequencing technology and bioinformatics analysis methods in the article, including steps involved in data analysis and specifics about the software used, ensuring the study's replicability.

2. Offer a more specific explanation of the steps involved in denoising and deduplication of sequencing data to ensure the accuracy and reliability of research results.

3. Provide a more thorough explanation, elucidating the ecological significance of diversity and abundance indices such as Chao1, Ace, Shannon, and Simpson. This will help readers better understand how these indices reflect the characteristics of microbial communities.

4. Provide detailed explanations of the statistical analysis methods used to characterize the microbial communities in different parts, including which statistical tests were employed and whether multiple comparison corrections were applied.

5. Include graphical representations of the results from Mothur software for diversity analysis to better illustrate the outcomes of diversity analysis.

6. The article mentions changes in microbial diversity and community structure without delving into the relationship between these changes and the physiological functions and health status of paddlefish. Further exploration through functional analysis is recommended to investigate the correlation between microbial composition and physiological functions such as nutrient absorption and the immune system in paddlefish.

7. The ariticle does not address whether the study considered temporal variations. Microbial communities may undergo changes over time. Conducting a time-series study would provide better insights into the dynamic changes of microbial communities and their relationship with seasonal or other time-related factors.

8. Analyze the metabolic functions of the microbial community to gain a deeper understanding of their roles in different parts of paddlefish. This includes studying functions such as enzyme production and acid production, which can help explain the microbial involvement in the digestive process of paddlefish.

9. Consider the microbial differences among individual paddlefish, taking into account factors such as age and gender. This in-depth analysis can provide insights into the individual variations in microbial communities among paddlefish.

Reviewer #2: The authors conducted a preliminary exploration of the microbiome in different segments of the digestive tract of paddlefish (Polyodon spathula). It is noteworthy that the experimental structure and design closely resemble the approach outlined by Yang G. et al., as published in 2020 (https://doi.org/10.1016/j.aqrep.2020.100402). While the study offers some insights, the novelty is somewhat lacking, and the microbiome analysis appears restricted to composition and abundance. Moreover, the content presented in tables and figures seems repetitive.

To enhance the manuscript, the authors should consider providing a more in-depth discussion of the findings, emphasizing unique aspects that distinguish their work from Yang G. et al. Additionally, addressing the following points will contribute to the overall clarity and rigor of the study:

Materials and Methods:

Line 100: Clarify the rationale for using fish within the weight range of 1967.2-2471.4g for the study.

Lines 112-115: Specify which part of the organ (e.g., saliva, mouth tissue, digesta, or intestine tissue) was utilized for microbiome analysis.

Line 148: Reconsider the use of T-test for comparing three groups of data. A more suitable approach, such as Kruskal-Wallis, followed by post hoc tests like Dunn’s test, should be employed.

Reviewer #3: This study investigated the microbial diversity and community structure in intestine, stomach, and mouth of paddlefish using high-throughput sequencing, which provide a microbiological basis for the development of aquaculture. However, there are still some problems worth further discussion and revision. Please refer to the attachment for details.

6. PLOS authors have the option to publish the peer review history of their article (what does this mean?). If published, this will include your full peer review and any attached files.

Reviewer #1: No

Reviewer #2: No

Reviewer #3: No

---

## [Author Response · Author response to Decision Letter 0]

7 Feb 2024

Dear PhD Nafiu Bala Sanda

Thank you for your letter and reviewers' constructive comments on our manuscript entitled "Microbiota in different digestive tract of paddlefish (Polyodon spathula) are related to their functions". Those comments are all valuable and helpful for revising and improving our paper. We have studied the comments carefully. According to the reviewers' detailed suggestions, we have made extensive revision on the original manuscript and below we present a point-by-point response to the comments.

With best wishes,

Sincerely yours

Chengxing Long 

Replies to comments from the Editors and Reviewers:

Comments

This study investigated the microbial diversity and community structure in intestine, stomach, and mouth of paddlefish using high-throughput sequencing, which provide a microbiological basis for the development of aquaculture. However, there are still some problems worth further discussion and revision. Details are given below. 

（1）The title should be revised, since the article only conducted microbial sequencing and did not measure its function.

Response: Thanks for the reasonable advice. I made a few revisions to the title. In this study, we mainly elaborate the relationship between different digestive tract microbial characteristics and their functions of the paddlefish. At the same time, in the discussion section, we added some materials explaining the relationship between different digestive tract microbes and their functions. Look forward to your further good advice.

（2）I don't quite understand: Line 100 “...with body weight 1967.2-2471.4 g... ” , but Line 102 “...the stocking specification was 350-400 g.”, what does it refer to?

Response: Thanks for the reasonable advice. 350-400 g is the weight of the paddlefish when it was first released into the pond. 1967.2-2471.4 g is the weight of the paddlefish when it was caught after 380 days of feeding. 

（3）The sampling time is only one, why does it take one year for breeding? Where is the meaning?

Response: Thanks for the reasonable advice. In order to better analyze the correlation between the microbiota in different parts of the digestive tract and the function of each digestive tract of paddlefish. We select the same batch of fish, feed the same pond, sample at the same time.

（4）There are only 5 samples, not enough.

Response: Thanks for the reasonable advice. Indeed, the larger the sample size, the more convincing it can be. However, due to the limitations of funds and conditions, we finally selected 5 healthy individuals of the same size from the captured fish for the experiment.

（5）The standard deviations of the table 2 and 3 data are very large, why?

Response: Thanks for the reasonable advice. The standard deviation of some species is indeed a bit large, but it is acceptable for species with higher abundance, and these results may be related to individual differences, or the low abundance of these species themselves.

（6）Line 202: “.644 genera were detected in YC of paddlefish. ”, should be “...644 genera were detected in YZ of paddlefish.”

Response: Thanks for the reasonable advice. Line 202: should be “...644 genera were detected in YZ of paddlefish.”

（7）Line 248-249: “These research results were the best evidence for the high body fat content in the paddlefish.” this sentence is inappropriate, so suggested to delete it.

Response: Thanks for the reasonable advice. The sentence has been deleted.

（8）Line 255-257: “Clostridium can antagonize a variety of pathogenic fungi and also promote nitrogen accumulation (Qian et al., 2018), which will provide evidence for the water quality optimization function of paddlefish.” this sentence is also not very accurate, it is recommended to modify it.

Response: Thanks for the reasonable advice. The sentence has been revised.

Journal Requirements:

Response: Thanks for the reasonable advice. It has been modified as required.

We noticed you have some minor occurrence of overlapping text with the following previous publication(s), which needs to be addressed:

Response: Thanks for the reasonable advice. In order to reduce the pain of the fish as much as possible, the paddlefish was deeply anesthetized in water containing overdose of MS222. After wiping the outer surface of the paddlefish with 75% ethanol, the contents of the mouth, stomach and intestine of the paddlefish were collected by sterile tools.

"the Natural Science Foundation of Hunan Province (2022JJ30316)."

Response: Thanks for the reasonable advice. The funder had no role in study design, data collection and analysis, decision to publish, or preparation of the manuscript, and added in the cover letter.

"This work was supported by the Natural Science Foundation of Hunan Province (2022JJ30316)"

"the Natural Science Foundation of Hunan Province (2022JJ30316)."

Response: Thanks for the reasonable advice. The fund information was removed from the Acknowledgments section.

Response: Thanks for the reasonable advice. The ORCID iD has been provided.

6. Please remove your figures from within your manuscript file, leaving only the individual TIFF/EPS image files, uploaded separately. These will be automatically included in the reviewers’ PDF.

Response: Thanks for the reasonable advice. Figures and tables in the manuscript have been removed.

Reviewer #1: 

1. Provide more detailed descriptions of the high-throughput sequencing technology and bioinformatics analysis methods in the article, including steps involved in data analysis and specifics about the software used, ensuring the study's replicability.

Response: Thanks for the reasonable advice. We have listed detailed references for the methods and software involved in the article, which can ensure the repeatability of the research.

2. Offer a more specific explanation of the steps involved in denoising and deduplication of sequencing data to ensure the accuracy and reliability of research results.

Response: Thanks for the reasonable advice. In order to ensure the accuracy and reliability of the research results, the steps involved in de-noising and de-duplication of sequencing data were explained in detail, and detailed references were noted. (p132-p137)

3. Provide a more thorough explanation, elucidating the ecological significance of diversity and abundance indices such as Chao1, Ace, Shannon, and Simpson. This will help readers better understand how these indices reflect the characteristics of microbial communities.

Response: Thanks for the reasonable advice. The ecological significance of the diversity and abundance indices such as Chao1, Ace, Shannon and Simpson has been well studied, so I will not elaborate in detail here, and I will add it if necessary.

4. Provide detailed explanations of the statistical analysis methods used to characterize the microbial communities in different parts, including which statistical tests were employed and whether multiple comparison corrections were applied.

Response: Thanks for the reasonable advice. Statistical methods have been supplemented. 

5. Include graphical representations of the results from Mothur software for diversity analysis to better illustrate the outcomes of diversity analysis.

Response: Thanks for the reasonable advice. I consider that the data in the table can illustrate the change in diversity more clearly than the graph, so the results of the diversity analysis are not represented by the graph. We look forward to your approval.

6. The article mentions changes in microbial diversity and community structure without delving into the relationship between these changes and the physiological functions and health status of paddlefish. Further exploration through functional analysis is recommended to investigate the correlation between microbial composition and physiological functions such as nutrient absorption and the immune system in paddlefish.

Response: Thanks for the reasonable advice. Exploring the correlation between microbial composition and physiological functions such as nutrient absorption and the immune system in paddlefish is the main content of my next research.

7. The ariticle does not address whether the study considered temporal variations. Microbial communities may undergo changes over time. Conducting a time-series study would provide better insights into the dynamic changes of microbial communities and their relationship with seasonal or other time-related factors.

Response: Thanks for the reasonable advice. You have provided me with a good research idea. In the following research, I will consider the relationship between dynamic changes of microbial communities and seasonal or other time-related factors.

8. Analyze the metabolic functions of the microbial community to gain a deeper understanding of their roles in different parts of paddlefish. This includes studying functions such as enzyme production and acid production, which can help explain the microbial involvement in the digestive process of paddlefish.

Response: Thanks for the reasonable advice. Analyze the metabolic functions of the microbial community to gain a deeper understanding of their roles in different parts of paddlefish. We will focus on this in the following research.

9. Consider the microbial differences among individual paddlefish, taking into account factors such as age and gender. This in-depth analysis can provide insights into the individual variations in microbial communities among paddlefish.

Response: Thanks for the reasonable advice. There are many factors affecting the differences in the microbial communities of paddlefish. In this study, we only considered the relationship between the digestive tracts of paddlefish and the water environment

Reviewer #2: 

Materials and Methods:

Line 100: Clarify the rationale for using fish within the weight range of 1967.2-2471.4g for the study.

Response: Thanks for the reasonable advice. The weight of the paddlefish is not controlled by us, and we only want to analyze the microbiota characteristics of different digestive tracts of the paddlefish after one year of feeding. The results of this study aim to show that the microbial characteristics are related to their function.

Lines 112-115: Specify which part of the organ (e.g., saliva, mouth tissue, digesta, or intestine tissue) was utilized for microbiome analysis.

Response: Thanks for the reasonable advice. The digestive system was analyzed in this study

Line 148: Reconsider the use of T-test for comparing three groups of data. A more suitable approach, such as Kruskal-Wallis, followed by post hoc tests like Dunn’s test, should be employed.

Response: Thanks for the reasonable advice. We have modified and supplemented the statistical methods.

Reviewer #3: 

 This study investigated the microbial diversity and community structure in intestine, stomach, and mouth of paddlefish using high-throughput sequencing, which provide a microbiological basis for the development of aquaculture. However, there are still some problems worth further discussion and revision. Please refer to the attachment for details.

Response: Thanks for the reasonable advice .We have made effective revised to the article in accordance with the requirements of reviewers and editors, and hope that you will continue to give valuable comments.

---

## [Editor Report · Decision Letter 1]

8 Apr 2024

Microbiota in different digestive tract of paddlefish (Polyodon spathula) are related to their functions.

PONE-D-23-38263R1

Dear Dr. Long Chengxing,

We’re pleased to inform you that your manuscript has been judged scientifically suitable for publication and will be formally accepted for publication once it meets all outstanding technical requirements.

Kind regards,

Nafiu Bala Sanda, PhD

Academic Editor

PLOS ONE

Additional Editor Comments (optional):

The manuscript can be accepted for publication in PLOS ONE journal. Congratulation!
---

## [Editor Report · Acceptance letter]

8 May 2024

PONE-D-23-38263R1 

PLOS ONE

Dear Dr. Long, 

I'm pleased to inform you that your manuscript has been deemed suitable for publication in PLOS ONE. Congratulations! Your manuscript is now being handed over to our production team.

Kind regards, 

on behalf of

Dr. Nafiu Bala Sanda 

Academic Editor

PLOS ONE